# Management of Postpartum Extensive Venous Thrombosis after Second Pregnancy

**DOI:** 10.3390/medicina59050871

**Published:** 2023-04-30

**Authors:** Andreea Taisia Tiron, Anca Filofteia Briceag, Liviu Moraru, Lavinia Alice Bălăceanu, Ion Dina, Laura Caravia

**Affiliations:** 1Faculty of Medicine, “Carol Davila” University of Medicine and Pharmacy, 050474 Bucharest, Romania; taisia_andreea@yahoo.com (A.T.T.); alicebalaceanu@yahoo.com (L.A.B.); ion.dina@umfcd.ro (I.D.); 2Department of Cardiology, ”St. John” Emergency Hospital, 13 Vitan Barzesti Street, 042122 Bucharest, Romania; briceag_anca@yahoo.ro; 3Department of Anatomy, ”George Emil Palade” University of Medicine, Pharmacy, Sciences and Technology, 540142 Targu Mures, Romania; 4Department of Internal Medicine, ”St. John” Emergency Hospital, 13 Vitan Barzesti Street, 042122 Bucharest, Romania; 5Department of Gastroenterology, ”St. John” Emergency Hospital, 13 Vitan Barzesti Street, 042122 Bucharest, Romania; 6Division of Cellular and Molecular Biology and Histology, Department of Morphological Sciences ”Carol Davila” University of Medicine and Pharmacy, 050474 Bucharest, Romania; laura.caravia@umfcd.ro

**Keywords:** postpartum, venous thrombosis, pregnancy, management

## Abstract

*Background*: Pregnancy induces a physiological prothrombotic state. The highest risk period for venous thromboembolism and pulmonary embolism in pregnant women is during the postpartum period. *Materials and Methods*: We present the case of a young woman who gave birth 2 weeks before admission and was transferred to our clinic for edema. She had an increased temperature in her right limb, and a venous Doppler of the limb confirmed thrombosis of the right femoral vein. From the paraclinical examination, we obtained a CBC with leukocytosis, neutrophilia, and thrombocytosis, and a positive D-dimer test. Thrombophilic tests were negative for AT III, lupus anticoagulant negative, and protein S and C, but were positive for heterozygous PAI-1, heterozygous MTHFR A1298C, and EPCR with A1/A2 alleles. After 2 days of UFH with therapeutic APTT, the patient had pain in her left thigh. We performed a venous Doppler, which revealed bilateral femoral and iliac venous thrombosis. During the computed tomography examination, we assessed the venous thrombosis extension on the inferior cava, common iliac, and bilateral common femoral veins. Thrombolysis was initiated with 100 mg of Alteplase given at a rate of 2 mg/h; however, this did not lead to a considerable reduction in the thrombus. Additionally, the treatment with UFH was continued under therapeutic APTT. After 7 days of UFH and triple antibiotic therapy for genital sepsis, the patient had a favorable evolution with remission of venous thrombosis. *Results*: Alteplase is a thrombolytic agent that is created with recombinant DNA technology, and it was successfully used to treat thrombosis that occurred in the postpartum period. *Conclusions*: Thrombophilias are associated with a high VTE risk but also with adverse pregnancy outcomes, including recurrent miscarriages and gestational vascular complications. In addition, the postpartum period is associated with a higher VTE risk. A thrombophilic status with heterozygous PAI-1, heterozygous MTHFR A1298C, and EPCR with A1/A2 positive alleles is associated with a high risk of thrombosis and cardiovascular events. Thrombolysis can be successfully used postpartum to treat VTEs. Thrombolysis can be used successfully in VTE developed in the postpartum period.

## 1. Introduction

Pregnancy induces a physiological prothrombotic state. The highest risk period for venous thromboembolisms is during the postpartum period [1]. Pregnancy increases the risk of thrombosis due to the hormonal, biological changes in the body. These changes occur in the blood flow via venous stasis from a pregnant uterus; changes in the vascular wall and coagulation factors (increased levels of factor II, V, VII, VIII, IX, X, von Willebrand factor); and increased D-dimer, fibrinogen, activation of platelets, and activity levels of natural anticoagulants (protein C, protein S, and plasminogen activator inhibitor-1). All these changes persist for another 6 weeks postpartum [2,3,4]. The VTE risk is higher in the third trimester, but the highest risk is observed postpartum. Thus, the relative risk is 20 times higher in the first six weeks postpartum, and 80% of these thrombotic events occur in the first three weeks postpartum [2]. Compared with the nonpregnant state, the 6-week postpartum period is associated with increases in stroke risk by a factor of 3 to 9, in myocardial infarction risk by a factor of 3 to 6, and in venous thromboembolism risk by a factor of 9 to 22 [5]. The risk can persist up to 12 weeks after delivery, but it is higher in the first 6 weeks. Guidelines recommend LMWH (Low Molecular Heparin) 6 weeks postpartum [6]. The obstetric complications associated with VTE are pregnancy loss, IUGR (Intrauterine Growth Retard), fetal death, abruptio placentae, and PE (Preeclampsia) [7,8]. Women experience five times more frequent DVT during pregnancy due to their hypercoagulability state, which is a defense mechanism against excess bleeding in the case of a miscarriage and childbirth [7,8]. In developing countries, postpartum hemorrhage is a frequent cause of death in pregnancy. The frequency of the risk of thrombosis increases in trimester III and postpartum, presenting the following clinical characteristics limb edema, skin changes, ulcerations, and recurrent thrombosis. Frequent associations with DVT are inherited thrombophilia, acquired thrombophilia, a history of thrombosis, heart disease, and sickle cell disease. To this, we add obesity, age over 35 years old, multiparity, nulliparity, immobilization in bed or sedentary lifestyle, smoking, assisted reproduction procedures, diabetes, infections, and caesarean sections. Virchov’s triad is an essential pathogenic mechanism as it involves venous stasis, endothelial damage, and hypercoagulability [3]. Antiphospholipid syndrome [APS] and autoimmune diseases are also responsible for recurrent thrombosis. APS promotes a hypercoagulation state during pregnancy. APS is an autoimmune disease characterized by the presence of antiphospholipid antibodies that promote recurrent arterial and venous thrombosis, which is accompanied by important complications during pregnancy; these complications are sometimes catastrophic and can lead to the failure of multiple organs [9]. Clarifying the pathogenic mechanism is essential when providing therapy for the disease [9]. The antibodies that must be tested to confirm the disease are the lupus anticoagulant, antibodies against cardiolipin, and antibodies against beta 2 glycoprotein [10]. They are sources of procoagulant cell activation. They activate the complement cascade, which leads to an increase in the capillary permeability, the activation of platelets and neutrophils, and the release of cytokines and TNF alpha from monocytes, which accelerate inflammation and trigger the coagulation cascade. They activate endothelial cells that produce prothrombotic molecules via the activating complement [11].

The management consists of administering anticoagulants, among which the preferred one is LMWH; with the necessary precautions, anticoagulants produce HIT syndrome with thrombocytopenia and are administered with caution in renal diseases, depending on molecular weight and the chain, being cleared from the body by the kidneys. Warfarin transplacentally passes, which causes fetal bleeding, neurological damage, stillbirths, and low intelligence, which is why it is not used during pregnancy. In the case of Direct Factor Xa inhibitors, we do not know their side effects during pregnancy [3].

Identifying patients at risk of thrombosis and using anticoagulant prophylaxis in those patients during pregnancy and 6 weeks postpartum is important. In total, 15% of the population has thrombophilia, and thrombophilia is associated with DVT in proportions up to 50% in certain conditions [3]. The most serious form is that with homozygous Leyden Factor V. Protein C and S deficiencies, as well as prothrombin G20210A or antithrombin deficiencies are also important. The American College of Chest Physicians recommends prophylaxis with LMWH in all pregnant women with a history of VTE and thrombophilia or in those with a history of more than two DVT episodes, without consensus regarding the dose. For those with thrombophilia without VTE, prophylaxis with LMWH is not recommended until postpartum [3,12,13]. In recent years, VTE risk has increased [14].

Pregnancy with thrombophilia induces a high-risk state for VTE, and sepsis only increases the risk. Assisted human reproduction methods also increase the VTE risk to a lesser extent [15]. A series of pathological conditions besides APS and hereditary thrombophilia can be associated with thrombotic risk: von Willebrand disease, sickle cell, inflammatory bowel disease, heart disease, smoking, immobilization, maternal age, obesity, and being African American. The recommended prophylaxis is LMWH [16,17].

Inherited bleeding disorders are pathological conditions that are also accompanied by VTE and determine mortality with a 6% prevalence [16]. The most common inherited bleeding disease is hemophilia, but it is very rare, not specific to pregnancy, and can appear in newborns who have thrombotic and hemorrhage disorders. In pregnancy, diagnosing bleeding disorders is difficult due to the lack of specific laboratory tests [18]. Thrombotic microangiopathy (TMA) consists of TTP (Thrombotic Thrombocytopenic Purpura) and HUS (Hemolytic Uremic Syndrome), which is associated with increased mortality and is difficult to diagnose [16]. Cardiac valvulopathies with valve prostheses, especially mechanical, are associated with a state of increased hypercoagulability, which requires management with a multidisciplinary team of cardiologists, obstetricians, anesthesiologists, neonatologists, and radiologists [16].

Thromboses are 75–82% venous and 20–25% arterial. VTEs occur during pregnancy in 0.5–2/1000 women. VTEs are responsible for 1.5 deaths per 100,000 deliveries in the USA [17].

In pregnant women, most thrombotic events occur in the left iliofemoral and inferior vena cava because of the increased venous stasis and compression by the pregnant uterus [5,19].

This manuscript is about the progression of thrombosis under LMWH treatment in a 20-year-old postpartum woman without prothrombotic risk factor and which evolved unexpectedly aggressively, so thrombolysis was necessary. 

## 2. Case Report

We present the case of a 20 year old woman giving birth 2 weeks before admission. This was her second pregnancy; her first pregnancy was carried out to term without complications, and she had no family history of VTEs and no other known risk factors. The delivery occurred without complications, and the baby had an Apgar score of 10. During the pregnancy, the patient gained 30 kg in weight. At the time of the delivery the risk of VTE was low and prophylaxis was not indicated.

Two weeks after the delivery, the patient was admitted to the gynecology ward for right limb edema with erythema and a local increased temperature. After conducting a venous Doppler evaluation, right femoral and popliteal vein thrombosis was confirmed. The patient was transferred to the cardiology department for treatment and further investigations.

On admission, she was hemodynamically stable, and at the physical exam, she presented with right limb erythema, edema, and warm skin, with a positive Homans sign and normal pulmonary murmur without lung crackles. Her respiratory rate was 22 breaths/min, and her SaO2 = 94%, and she had a normal cardiac rhythm with no added heart sounds, an arterial blood pressure of 110/60 mmHg, a heart rate of 76 bpm, and no other remarkable findings on the clinical exam. The gynecological examination revealed a normal appearance and physiological involution of the uterus without signs of bleeding.

From the paraclinical examination, we obtained a CBC with leukocytosis, neutrophilia, WBC = 14.71 × 10^3^/µL and NEU = 11.61 × 10^3^/µL, anemia with Hb = 11.5 g/dL, HT = 35%, MCH = 24.63 pg, thrombocytosis with PLT = 439 × 10^3^/µL, CRP = 35.42 mg/L, fibrinogen = 558.99 mg/dL, and D-dimer = 760 µg/mL. The thrombophilic tests were conducted with AT III in normal ranges, a negative lupus anticoagulant, and S and C proteins in normal ranges. Following the investigation for hereditary thrombophilia, the tests were positive for the heterozygous PAI-1, MTHFR A1298C, EPCR with A1/A2 alleles, and homozygous 4G/4G genotype, which corresponds to minor thrombophilia. These associations decrease the plasma levels of inhibitor activator plasminogen and the fibrinolytic activity with a high thrombosis risk. With this new information, adding sepsis and thrombophilia, the patient’s risk score for VTE increased to high risk. The results of the ECG at admission demonstrated the following: a sinus rhythm with 130 bpm and an QRS axis +60 degrees without other changes in the ST segment (Figure 1). 

Echocardiography was performed, and the results revealed normal cardiac cavities, a normal LV ejection fraction, and diastolic dysfunction grade I without valvular dysfunction or pericardial effusion.

In our clinic, the patient received treatment with UFH (Unfractioned Heparin), therapeutic APTT, and analgesics for 2 days, and she experienced no reduction in the edema of her limbs. The patient was treated with heparin to treat an intense pain in her left thigh, and a venous Doppler was performed, which confirmed the bilateral iliac and femoral vein thrombosis. We decided to initiate thrombolysis on a central venous catheter with Alteplase, which did not lead to a considerable reduction in the thrombus, followed by UHF therapy by using therapeutic APTT.

The results of the abdominal and pelvic computed tomography examination revealed venous thrombosis in the inferior cava vein with an extension to both the common iliac veins and internal and external iliac veins, and it also extended to the bilateral femoral veins (Figure 2). The pulmonary artery and its branches were completely opacified without signs of pulmonary thromboembolism. Additionally, minimal basal right pleural fluid was observed.

Her general condition was worsening with important inflammatory syndrome and important leukocytosis with neutrophilia, and the anemia was also worsening with Hb = 9.6 g/dL, HT = 30.9%, MCH = 23.83 pg, MCV = 76.52 fL, and low seric iron at 13 ug/dL. We combined biological cultures with negative blood cultures and a positive culture from pharyngeal and nasal exudate to treat the Pseudomonas. Antibiotic therapy with 1 g/8 h of meropenem, 2 g/24 h of vancomycin, and 2 g/24 h of metronidazole was administered for 10 days to treat genital sepsis, and the patient experienced clinical improvements.

Post-thrombolysis, the patient received UFH for 7 days under the therapeutic control of APTT, triple antibiotic therapy, beta blockers, gastric antisecretory, and parenteral hydration, which resulted in a favorable evolution and a remission of the limb edema and inflammatory syndrome. The patient was discharged with an order to undergo an anticoagulant treatment of 15 mg of Rivaroxaban twice a day for 14 days and 20 mg/twice a day for the antiaggregant therapy. During the follow-up 2 months later, we found that the patient had partial right superficial vein thrombosis without other thromboses, but due to a genetic mutation, the patient was advised to take long-term oral anticoagulation medications. After long-term follow-up, at the Doppler ultrasound one year later, the partial right superficial vein thrombosis persisted. In this situation, it is recommended to continue the anticoagulation with aspenter and take into account the other procoagulant pathologies that could be triggered during a new pregnancy.

## 3. Discussion

The thrombophilia, postpartum status, and sepsis were the factors that induced DVT in our young patient after her second pregnancy. 

The highest risk period for venous thromboembolisms and pulmonary embolisms in pregnant women is during the postpartum period. Any prophylaxis against these events should be particularly targeted for postpartum women. Risk factors to be considered include prior VTE, familial VTE history, the presence of known thrombophilia, caesarean delivery, prolonged antepartum immobilization, increased body mass index (BMI), considerable pregnancy complications, and medical comorbidities [19].

Data, largely from retrospective cohorts and case–control studies, have shown that inherited thrombophilias are not only associated with VTE but also with adverse pregnancy outcomes, including recurrent miscarriage and gestational vascular complications. The most common inherited thrombophilias are Factor V Leiden (FVL) and Factor II (prothrombin) G20210A, which affect 3–11% of the population; less prevalent (<1%) are inherited thrombophilias, including protein C, protein S, an antithrombin deficiency, dysfibrinogenemia, and hyperhomocysteinemia [12,20,21]. 

A thrombophilic status with heterozygous PAI-1 and MTHFR A1298C and EPCR with A1/A2 positive alleles together are associated with decreasing plasma levels of inhibitor activator plasminogens and a decrease in fibrinolytic activity with a high thrombosis risk [22].

The results of prospective and retrospective studies have demonstrated a modest association of homocysteine with venous thrombosis, but when associated with other mutations such as PAI-1 heterozygous, homocysteine increases the cardiovascular and VTE risks. Heparins, UFH, and LMWH are the preferred agents for anticoagulation in pregnancy because they show no transplacental passage. Both heparins and warfarin are safe for the infant during breastfeeding. Under DOACs (Direct acting oral anticoagulants), breastfeeding is not recommended as no evidence of its safety exists [23].

Because anticoagulant treatment is proposed in 91% of cases, the preferred treatment would be LMWH as it has a strong safety profile, does not cross the placenta, and lowers the risk of preeclampsia; intra-arterial thrombolysis is used in 20% of cases, and a thrombectomy with thrombolysis is used in 8% of cases, with excellent outcomes if the diagnosis is correct [13,24]. DVT in pregnancy is associated with high morbidity and mortality. Pulmonary embolisms cause maternal death. DVT leads to long-term complications. Women develop DVT five times more frequently during pregnancy. Apart from the standard treatment, which is anticoagulation, an endovascular inferior cava filter can be installed, and the pharmacomechanical catheter for directed thrombolysis is only used in pregnancy or in the case of massive life-threatening situations as it has been associated with abruptio placentae, premature birth, and fetal damage [3,4]. When administering LMWH, antiFactor Xa dosage is not necessary in pregnancy and coumarins are contraindicated, but elastic compression is beneficial [4,13]. In the case of APS resistant to treatment, new therapeutic strategies should also be searched [25].

The signs used to provide a DVT diagnosis are swelling of the legs, discomfort, erythema, abdominal or pelvic pain, edema, and dyspnea [3,4]. D dimers are not measured because they are increased in pregnancy anyway. The diagnosis consists of imaging by CT, MRI, and a color Doppler with intraluminal echoes [3,26]. Compression duplex ultrasonography is also indicated in deep thrombosis and pulmonary imaging for possible pulmonary embolisms, and EKG shows tachycardia and nonspecific features that suggest pulmonary embolisms, oxygen saturation, or angiography in nonpregnant/postpartum patients, in addition to the leg circumference being 2 cm larger [4]. A Doppler velocimetry is essential in fetal assessment [25]. 

In inherited bleeding disorders, the management, as well as the diagnosis, is more complicated. Two types of TMA thrombotic microangiopathies exist: Thrombotic thrombocytopenic purpura (TTP) and hemolytic uremic syndrome (HUS). They can appear during pregnancy or postpartum. TTP can be acquired or can be a congenital deficiency of ADAMTS 13, and this especially occurs in the second and third trimester of pregnancy. The treatment consists of restoring ADAMTS 13 activity with plasmapheresis and immunosuppressive agents [27]. A HUS atypical complement is triggered by pregnancy. The treatment consists of the anti-C5 humanized monoclonal antibody eculizumab. We found that both situations were associated with pre-eclampsia and HELLP syndrome. We identified that the von Willebrand cleaving protease ADAMTS 13 is a major factor in TTP pathogenesis and clinically determined neurological damage, renal failure, and thrombosis. In HUS, we found that the dysregulation of the alternate complement pathway was a mechanism of HUS occurrence, and in this case, ADAMTS 13 was over 10% [18,27].

Only a few available sensitive laboratory tests exist, and the current ones are insufficient to make a diagnosis. This results in diagnostic challenges, problems in the current treatment, and the need for guidelines for more efficient management [11]. In some pregnancy complications associated with thrombosis or hemorrhage, the proteins that control blood clotting become overactive, and DIC sets in. From this point of view, the DIC disseminated intravascular coagulation score would be useful in predicting uncontrolled coagulopathy [18]. It consists of determining the complete blood cell count, prothrombin time (PT), partial thromboplastin time (APTT), fibrinogen, and D dimers. However, the DIC score is modified during pregnancy and cannot be used because fibrinogen increases, platelets decrease, and the PT and PTT do not change during pregnancy. Only modified PTs are used as DIC indicators. DIC indicators are present during uncontrolled bleeding at birth, abruptio placentae, placenta previa, uterine rupture, cervical and vaginal laceration, infections, surgical techniques, and HELLP syndrome. Corrected by surgery and blood transfusions, DIC is associated with thrombocytopenia below 50,000, prolonged PT, increased dimers, and low fibrinogen. Additionally, an algorithm that compares the types of bleeding and thrombosis that can appear in pregnancy with the laboratory tests specific to each disease would also be useful in terms of making life-saving diagnoses and management [18].

Von Willebrand disease seems to affect 0.01–1.3% of the general population, and it is the most common inherited mild bleeding condition. Because a high risk of bleeding is present at birth, F VIII must be restored. Blood-product administration is performed to manage PPH, and therefore it requires massive transfusion protocols, laboratory monitoring, and a multidisciplinary approach [16].

Pregnancy failure is prevented by conventional therapy in women with APS. Various treatment strategies are leading to remarkably higher live birth rates. Conventional therapy consists of a prophylactic dose of heparin plus a low dose of aspirin to prevent pregnancy loss, and for those with a history of thrombosis, a therapeutic dose of heparin with LDA is provided instead. Here, we add intravenous immunoglobulin, low-dose prednisolone, plasmapheresis, and immunoadsorption. New clinical studies are needed. During diagnosis, in addition to antibody dosing, a Doppler velocimetry showing notch on the uterine arteries suggests a poor prognosis. The activation of complement cascades and inflammation are involved in trophoblast invasion disorder [28]. An innovative treatment would be the slow infusion of the tissue-type plasminogen activator in the case of thrombosed mechanical valves [29]. A risk of hemorrhage and preterm birth exists.

Choosing contraceptives after venous thromboembolism (VTE) is challenging because hormonal contraceptives may increase the risk of recurrent VTE. Estrogen contraceptives are usually contraindicated in women with a personal VTE history, and the use of oral progestin-only contraceptives is still being studied. Nonhormonal contraceptives are recommended [30]. For the next pregnancy of this patient, and to reduce the associated risks, we recommend using LMWH after conception.

A multidisciplinary team that includes a cardiologist, hematologist, and gynecologist is required to prevent thrombotic events in the future, especially if a third pregnancy is desired, as her VTE risk score is high, even in the absence of an infection. Considering the long-term follow-up with the presence of residual thrombosis at such a young age with minor thrombophilia, which does not explain the dramatic evolution of VTE that appeared in this case, we suggest future investigation of all conditions presented here as possible procoagulant pathologies.

## 4. Conclusions

Thrombophilias are associated with a high VTE risk but also with adverse pregnancy outcomes, and the postpartum period is associated with a higher VTE risk. One’s thrombophilic status is associated with a high risk of thrombosis and cardiovascular events in the postpartum period. Thrombolysis can be successfully used in this situation, and Alteplase is a thrombolytic agent that is created with recombinant DNA technology and was effective in this case, along with LMWH.

## Figures and Tables

**Figure 1 medicina-59-00871-f001:**
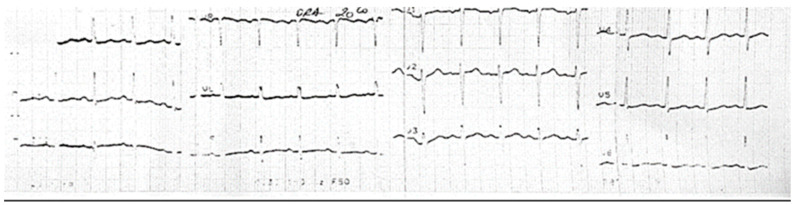
Sinus tachycardia, no pathological changes.

**Figure 2 medicina-59-00871-f002:**
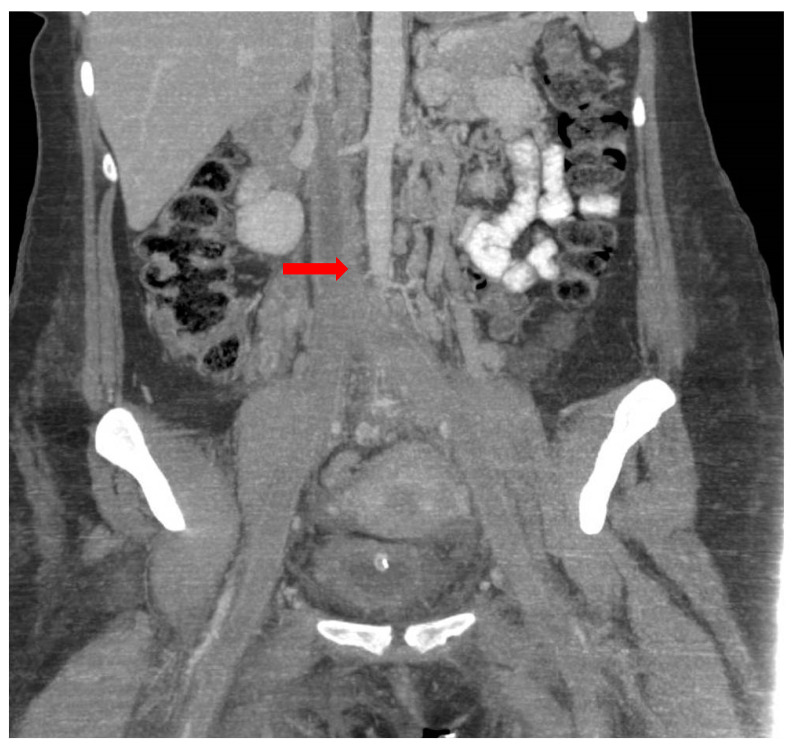
Venous thrombosis in inferior cava vein with extension to the common iliac veins and bilateral femoral veins (red arrow).

## Data Availability

Data supporting reported results can be found in the Department of Cardiology database, ”St. John” Emergency Hospital, 13 Vitan Barzesti Street, 042122 Bucharest, Romania.

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
