# Peer review of "Management of Postpartum Extensive Venous Thrombosis after Second Pregnancy"

_medicina, 2023, doi:10.3390/medicina59050871_

Round 1

Reviewer 1 Report

This report describes a rare form of thrombophilia. But the structure of the paper and the English language need to be completely revised. The focus of the discussion should be clear. It is suggested that the author could summarize the focus and main points (highlights) of the article.本报告描述了一种罕见的血栓形成倾向。但是论文的结构和英语需要完全修改。讨论的重点应该明确。建议作者能总结出文章的重点和要点(亮点)。

“Thombolysis can be used successfully used postpartum in VTE.” Redundant “used”.“血栓溶解术可成功用于产后VTE.”多余的“使用”。

“The frequency of the risk of thrombosis increases in trimester III and lauzia, with limb edema, skin changes, ulcerations, recurrent thrombosis.” What does “lauzia” mean? I suggest that the authors revise this sentence for clarity.“在妊娠三期和lauzia期,血栓形成的风险增加,伴有肢体水肿,皮肤改变,溃疡,复发性血栓形成。”“lauzia”是什么意思?我建议作者修改这句话以使其清晰。

“15% of the population has thrombophilia and it is associated with DVT in proportion up to 50%.” The proportion is high. I suggest that references should be added here.“15%的人群有血栓形成倾向,与DVT相关的比例高达50%。”比例很高。我建议在这里加上参考资料。

“A 20-year-old woman gave biologically assisted birth 2 weeks before admission.” Does this mean pregnancy after assisted reproductive technology?“一名20岁的女性在入院前2周进行了生物辅助分娩。”这是否意味着辅助生殖技术后怀孕?

“At the time of the delivery calculating the risk of VTE with Geneva score she had a low risk and prophylaxis was not indicated.” Is the Geneva score for pregnant women? If there is no further explanation, references are required here.“在分娩时,用日内瓦评分计算VTE的风险,她的风险很低,没有预防措施。”日内瓦评分是孕妇的吗?如果没有进一步的解释,这里需要参考。

“fibrinogen=558,99 mg/dl, and D-dimer test was positive.” 558.99? Similar use of commas should be corrected. Can the number of the D-dimer be given here?“纤维蛋白原= 558,99 mg/dl,D-二聚体试验阳性.”五百五十八块九毛九类似的逗号用法应予以纠正。D-二聚体的数量可以在这里给出吗?

“With this new information, adding sepsis and thrombophilia, the patient`s risk score for VTE increases to high risk.” The diagnostic basis for sepsis and thrombophilia should be described.“有了这个新的信息,加上败血症和血栓形成倾向,患者的VTE风险评分增加到高风险。应描述脓毒症和血栓形成倾向的诊断依据。

“followed by UHF therapy under therapeutic APTT.” UFH?“然后在治疗性APTT下进行超高频治疗”和睦家?

“TMA thrombotic microangiopathies are of two types” What is TMA?“TMA血栓性微血管病有两种类型”什么是TMA?

“From this point of view, the DIC score would be useful in predicting uncontrolled coagulopathy.” Why is DIC discussed in this paragraph? Seems unrelated to the case? Some explanation needs to be given here.“从这个角度来看,DIC评分在预测不受控制的凝血病方面是有用的。”为什么在本段中讨论DIC?似乎和案子无关?这里需要作一些解释。

Author Response

Dear reviewer,

We highlight here the main points of the article, being a case report : abstract, introduction with important diseases that are associated with an increased risk of thrombosis in pregnancy , case report ,discussion and conclusion, as suggested in the templates. To make things clearer regarding our decision to present the most important diseases with procoagulant potential , in line 212-224 we added :” Post thrombolysis the patient receives 7 days of UFH under therapeutic control of APTT, triple antibiotic therapy, beta blocker, gastric antisecretory, parenteral hydration, with favorable evolution, remission of limb edema and remission of the inflammatory syndrome. The patient is discharged with anticoagulant treatment of Rivaroxaban 15mg twice a day for 14 days, and 20 mg/twice a day, antiaggregant therapy. On follow-up 2 months later, the patient had partial right superficial vein thrombosis without other thrombosis, but due to genetic mutation the patient is advised for long term oral anticoagulation. After long-term follow-up , at the Doppler ultrasound one year later the partial right superficial vein thrombosis persists. In this situation it is recommended to continue the anticoagulation with aspenter and take into account the other procoagulant pathologies that could be triggered during a new pregnancy.” In addition, the progression of thrombosis under LMWH treatment in a 20-year-old woman without prothrombotic risk factor was unexpectedly aggressive, so thrombolysis was necessary (line 128-129). In discussion also we added line 337-343:” Multidisciplinary team with cardiologist, hematologist and gynaecologist is required to prevent thrombotic events in future, especially if a third pregnancy is desired, her VTE risk score being high even in the absence of infection. Considering the long term follow -up with the presence of residual thrombosis at such a young age with minor thrombophilia which does not explain the dramatic evolution of VTE that appeared in this case we suggest future investigation of all conditions presented here as possible procoagulant pathologies”.

We rephrased in the abstract line 48: ”Thombolysis can be used successfully in VTE developed in the postpartum period.”

We revised the sentence: ”The frequency of the risk of thrombosis increases in trimester III and postpartum, presenting the following clinical characteristics limb edema, skin changes, ulcerations, recurrent thrombosis.” (line 74-75)

We added reference and we revised the sentence: ”15% of the population has thrombophilia andin these population with thombophilia DVT is associated in proportion up to 50% in certain conditions[3]” (line 107-108)

No , she did not obtained pregnancy after ART and we changed the sentence : ”We present the case of 20 years old women giving birth 2 weeks before admission.” (line 146)

One of our colleagues used a criteria that is no longer used, we removed the sentence.

We corrected the commas, thank you and D-dimer=760 µg/mL - line 170

We supported the diagnosis in line 173-174: the tests were positive for heterozygous PAI-1, MTHFR A1298C, EPCR with A1/A2 alleles, homozygous 4G/4G genotype which corresponds to minor thrombophilia; and line 204-205 important leukocytosis with neutrophilia, line 208-210 positive culture from pharyngeal and nasal exudate for Pseudomonas.

UFH means Unfractioned Heparin - line187.

Thrombotic microangiopathy (TMA) consists of TTP (Thrombotic Thrombocytopenic Purpura) and HUS (Hemolytic Uremic Syndrome) - line 128-129, 280-281.

In some pregnancy complications associated with thrombosis or hemorrhage the proteins that control blood clotting become overactive and DIC sets in. From this point of view, the DIC disseminated intravascular coagulation score would be useful in predicting uncontrolled coagulopathy [18]. - line 297-299.

Thank you.

Reviewer 2 Report

- Line 63:  "Among the LMWHs Fraxiparine (Nadroparin 63 calcium) is not associated with HIT syndrome with thrombocytopenia" I am very surprised of this statement as I checked my local SmPC and find a risk of HIT in the warning section. I did not find Mitranovici on medline and Othman publication is describing the registries not the results... Can I ask you to provide another reference that could be superior to the SmPC or to reconsider this statement? 

-Line 89: "LMWH, with the necessary precautions, they produce HIT syndrome, 89 with thrombocytopenia and are administered with caution in renal diseases, being cleared 90 from the body by the kidneys". I agree with the caution, but there are not all the same regarding renal excretion. Indeed, the MW and the chains (long vs short) are +++ important. Please rephrase (Carrier M. Curr Oncol 2021, table 2)

- Line 93: "In the case of Antifactor X". By the book, antifactor X are including many drugs, including LMHWs. if you refer to the DOACs, I would suggest to use "Direct Factor Xa inhibitors", that would include Rivaroxaban, Apixaban, Edoxaban and Betrixaban but would exclude Dabigatran, VKAs and LMWHs.

- Line 227: "anti-factor X". Did you mean "anti-Factor Xa"? Are you tracking the activated one? or the non activated one? 

Author Response

Dear reviewer,

Line 63: Mitranovici is in Web of Science and PubMed, but you are right, we can not use only one reference to emphasize that the Fraxiparine is not associated with HIT syndrome, so I removed the sentence.

Line 99: We rephrased : LMWH are administered with caution in renal diseases, depending on molecular weight and the chain, being cleared from the body by the kidneys, accordingly with your suggestion.

Line 103: We used Direct FactorXa inhibitors.

Line 267: We refer to anti-Factor Xa , the activated form.

 Thank you.

Reviewer 3 Report

In the manuscript entitled „Management of postpartum extensive venous thrombosis after second pregnancy” the authors showed that thrombophilia is associated with a high risk of VTE but also with adverse pregnancy outcomes including recurrent miscarriage and gestational vascular complications.

The topic is very interesting, and the authors showed some interesting results. However, I have some suggestions for revision:

1.      It is necessary to revise the English language.

2.      Also, it is necessary to present the clinical course of the patient in more detail, including therapy and diagnostics.

Author Response

Dear reviewer,

  1. For English revision, we used MDPI services.
  2. The symptoms are extensively presented line 154-155, 159-166, laboratory tests 167-174,172-174. Then EKG,Doppler ultrasound, echocardiography line 175-186 support the diagnosis. The treatment is detailed line 187-194, the evolution of the case is presented line 195-200, CT-scan showed the evolution of thrombosis, figure 2 with description. Then we added recommendation: ”Post thrombolysis, the patient receives 7 days of UFH under therapeutic control of APTT, triple antibiotic therapy, beta blocker, gastric antisecretory, parenteral hydration, with favorable evolution, remission of limb edema and remission of the inflammatory syndrome. The patient is discharged with anticoagulant treatment of Rivaroxaban 15mg twice a day for 14 days, and 20 mg/twice a day, antiaggregant therapy. On follow-up 2 months later, the patient had partial right superficial vein thrombosis without other thrombosis, but due to genetic mutation, the patient is advised for long-term oral anticoagulation. After long-term follow-up, at the Doppler ultrasound one year later the partial right superficial vein thrombosis persists. In this situation it is recommended to continue the anticoagulation with aspenter and take into account the other procoagulant pathologies that could be triggered during a new pregnancy.” line 212-224.

For a better understanding of the case and its severity, we added to the discussion:” Multidisciplinary team with cardiologist, hematologist and gynaecologist is required to prevent thrombotic events in future, especially if a third pregnancy is desired, her VTE risk score being high even in the absence of infection. Considering the long-term follow-up with the presence of residual thrombosis at such a young age with minor thrombophilia which does not explain the dramatic evolution of VTE that appeared in this case we suggest future investigation of all conditions presented here as possible procoagulant pathologies.” line 337-343.

Thank you.

Round 2

Reviewer 1 Report

The previous issues have been addressed by the authors. Congratulations.
作者已经解决了前面的问题。恭喜你

Reviewer 2 Report

Ok for me